# Pervasive duplication of tumor suppressors in Afrotherians during the evolution of large bodies and reduced cancer risk

Juan M Vazquez[1†], Vincent J Lynch[2]*

[1]Department of Human Genetics, The University of Chicago, Chicago, United States; [2]Department of Biological Sciences, University at Buffalo, Buffalo, United States

*For correspondence:
vjlynch@buffalo.edu

Present address: [†] Department of Integrative Biology, University of California – Berkeley, Berkeley, United States

Competing interests: The authors declare that no competing interests exist.

**Abstract** The risk of developing cancer is correlated with body size and lifespan within species. Between species, however, there is no correlation between cancer and either body size or lifespan, indicating that large, long-lived species have evolved enhanced cancer protection mechanisms. Elephants and their relatives (Proboscideans) are a particularly interesting lineage for the exploration of mechanisms underlying the evolution of augmented cancer resistance because they evolved large bodies recently within a clade of smaller-bodied species (Afrotherians). Here, we explore the contribution of gene duplication to body size and cancer risk in Afrotherians. Unexpectedly, we found that tumor suppressor duplication was pervasive in Afrotherian genomes, rather than restricted to Proboscideans. Proboscideans, however, have duplicates in unique pathways that may underlie some aspects of their remarkable anti-cancer cell biology. These data suggest that duplication of tumor suppressor genes facilitated the evolution of increased body size by compensating for decreasing intrinsic cancer risk.

## Introduction

Among the constraints on the evolution of large bodies and long lifespans in animals is an increased risk of developing cancer. If all cells in all organisms have a similar risk of malignant transformation and equivalent cancer suppression mechanisms, then organisms with many cells should have a higher prevalence of cancer than organisms with fewer cells, particularly because large and small animals have similar cell sizes (*Savage et al., 2007*). Consistent with this expectation there is a strong positive correlation between body size and cancer incidence within species; for example, cancer incidence increases with increasing adult height in humans (*Million Women Study collaborators et al., 2011*; *Nunney, 2018*) and with increasing body size in dogs, cats, and cattle (*Dobson, 2013*; *Dorn et al., 1968*; *Lucena et al., 2011*). There is no correlation, however, between body size and cancer risk between species; this lack of correlation is often referred to as 'Peto's Paradox' (*Caulin and Maley, 2011*; *Leroi et al., 2003*; *Peto et al., 1975*). Indeed, cancer prevalence is relatively stable at ~5% across species with diverse body sizes ranging from the minuscule 51 g grass mouse to the gargantuan 4800 kg African elephant (*Abegglen et al., 2015*; *Boddy et al., 2020*; *Tollis et al., 2020*). The ultimate resolution to Peto's Paradox is trivial, large-bodied and long-lived species evolved enhanced cancer protection mechanisms, but identifying and characterizing the mechanisms that underlie the evolution of augmented cancer protection has proven difficult (*Ashur-Fabian et al., 2004*; *Seluanov et al., 2008*; *Gorbunova et al., 2012*; *Tian et al., 2013*; *Sulak et al., 2016*).

**eLife digest** From the gigantic blue whale to the minuscule bumblebee bat, animals come in all shapes and sizes. Any species can develop cancer, but some are more at risk than others. In theory, if every cell has the same probability of becoming cancerous, then bigger animals should get cancer more often since they have more cells than smaller ones. Amongst the same species, this relationship is true: taller people and bigger dogs have a greater cancer risk than their smaller counterparts.

Yet this correlation does not hold when comparing between species: remarkably large creatures, like elephants and whales, are not more likely to have cancer than any other animal. But how have these gigantic animals evolved to be at lower risk for the disease?

To investigate, Vazquez and Lynch compared the cancer risk and the genetic information of a diverse group of closely related animals with different body sizes. This included elephants, woolly mammoths and mastodons as well as their small relatives, the manatees, armadillos, and marmot-sized hyraxes. Examining these species' genomes revealed that, during evolution, elephants had acquired extra copies of 'tumour suppressor genes' which can sense and repair the genetic and cellular damages that turn healthy cells into tumours. This allowed the species to evolve large bodies while lowering their risk of cancer.

Further studies could investigate whether other gigantic animals evolved similar ways to shield themselves from cancer; these could also examine precisely how having additional copies of cancer-protecting genes helps reduce cancer risk, potentially paving the way for new approaches to treat or prevent the disease.

---

One of the challenges for discovering how animals evolved enhanced cancer protection mechanisms is identifying lineages in which large-bodied species are nested within species with small body sizes. Afrotherian mammals are generally small-bodied, but also include the largest extant land mammals. For example, maximum adult weights are ~70 g in golden moles, ~120 g in tenrecs, ~170 g in elephant shrews, ~3 kg in hyraxes, and ~60 kg in aardvarks (*Tacutu et al., 2013*). In contrast, while extant hyraxes are relatively small, the extinct Titanohyrax is estimated to have weighed ~1300 kg (*Schwartz et al., 1995*). The largest living Afrotheria are also dwarfed by the size of their recent extinct relatives: extant sea cows such as manatees are large bodied (~322–480 kg) but are relatively small compared to the extinct Stellar's sea cow which is estimated to have weighed ~8000–10,000 kg (*Scheffer, 1972*). Similarly African Savannah (4800 kg) and Asian elephants (3200 kg) are large, but are dwarfed by the truly gigantic extinct Proboscideans such as Deinotherium (~12,000 kg), *Mammut borsoni* (16,000 kg), and the straight-tusked elephant (~14,000 kg) (*Larramendi, 2015*). Remarkably, these large-bodied Afrotherian lineages are nested deeply within small-bodied species (*Figure 1*; *O Leary et al., 2013a*; *Springer et al., 2013*; *O Leary et al., 2013b*; *Puttick and Thomas, 2015*), indicating that gigantism independently evolved in hyraxes, sea cows, and elephants (Paenungulata). Thus, Paenungulates are an excellent model system in which to explore the mechanisms that underlie the evolution of large body sizes and augmented cancer resistance.

## Box 1. Eutherian phylogenetic relationships.

Eutheria (eu- 'good' or 'right' and thēríon 'beast', hence 'true beasts') is one of three living (extant) mammalian lineages (Monotremes, Marsupials, and Eutherians) that diverged in the early–late Cretaceous. Eutheria was named in 1872 by Theodore Gill and refined by Thomas Henry Huxley in 1880. Living Eutherians are comprised of 18 orders, divided into two major clades (*Figure 1A*): Atlantogenata including the superorders Xenarthra (armadillos, anteaters, and sloths) and Afrotheria (Proboscidea, Sirenia, Hyracoidea, Tublidentata, Afroinsectivora, Cingulata, and Pilosa), and Boreoeutheria including the superorders Laurasiatheria (Insectivora, Artodactyla, Pholidota, and Carnovora) and Euarchontoglires (Lagomorpha, Rodentia, Scandentia, Dermoptera, and Primates). In our analyses, we have focused on identifying gene duplications in Afrotherian and Xenarthran genomes (*Figure 1B*), using the Xenarthrans Hoffmans two-toed sloth (*Choloepus hoffmanni*) and nine-banded armadillo (*Dasypus*

*novemcinctus*) as out-groups to the Afrotherians. This approach allows us to use phylogenetic methods to polarize gene duplication events and identify genes that duplicated in the Afrotherian stem-lineage.

Many mechanisms have been suggested to resolve Peto's paradox, including a decrease in the copy number of oncogenes, an increase in the copy number of tumor suppressor genes (*Caulin and Maley, 2011*; *Leroi et al., 2003*; *Nunney, 1999*), reduced metabolic rates, reduced retroviral activity and load (*Katzourakis et al., 2014*), and selection for 'cheater' tumors that parasitize the growth of other tumors (*Nagy et al., 2007*), greater sensitivity of cells to DNA damage (*Abegglen et al., 2015*; *Sulak et al., 2016*), enhanced recognition of neoantigens by T cells, among many others. Among the most parsimonious routes to enhanced cancer resistance may be through an increased copy number of tumor suppressors. For example, transgenic mice with additional copies of *TP53* have reduced cancer rates and extended lifespans (*García-Cao et al., 2002*), suggesting that changes in the copy number of tumor suppressors can affect cancer rates. Indeed, candidate genes studies have found that elephant genomes encode duplicate tumor suppressors such as *TP53* and *LIF* (*Abegglen et al., 2015*; *Sulak et al., 2016*; *Vazquez et al., 2018*) as well as other genes with putative tumor suppressive functions (*Caulin et al., 2015*; *Doherty and de Magalhães, 2016*). These studies, however, focused on a priori candidate genes; thus it is unclear whether duplication of tumor suppressor genes is a general phenomenon in the elephant lineage or reflects an ascertainment bias.

Here we trace the evolution of body mass, cancer risk, and gene copy number variation across Afrotherian genomes, including multiple living and extinct Proboscideans (*Figure 1*), to investigate whether duplications of tumor suppressors coincided with the evolution of large body sizes. Our estimates of the evolution of body mass across Afrotheria show that large body masses evolved in a stepwise manner, similar to previous studies (*O Leary et al., 2013a*; *Springer et al., 2013*; *O Leary et al., 2013b*; *Puttick and Thomas, 2015*) and coincident with dramatic reductions in intrinsic cancer risk. To explore whether duplication of tumor suppressors occurred coincident with the evolution of large body sizes, we used a genome-wide Reciprocal Best BLAT Hit (RBBH) strategy to identify gene duplications and used maximum likelihood to infer the lineages in which those duplications occurred. Unexpectedly, we found that duplication of tumor suppressor genes was common in Afrotherians, both large and small. Gene duplications in the Proboscidean lineage, however, were uniquely enriched in pathways that may explain some of the unique cancer protection mechanisms observed in elephant cells. These data suggest that duplication of tumor suppressor genes is pervasive in Afrotherians and preceded the evolution of species with exceptionally large body sizes.

## Results

### Step-wise evolution of body size in Afrotherians

Similar to previous studies of Afrotherian body size (*Puttick and Thomas, 2015*; *Elliot and Mooers, 2014*), we found that the body mass of the Afrotherian ancestor was inferred to be small (0.26 kg, 95% CI: 0.31–3.01 kg) and that substantial accelerations in the rate of body mass evolution occurred coincident with a 67.36× increase in body mass in the stem-lineage of Pseudoungulata (17.33 kg); a 1.45× increase in body mass in the stem-lineage of Paenungulata (25.08 kg); a 11.82× increase in body mass in the stem-lineage of Tethytheria (296.56 kg); a 1.39× increase in body mass in the stem-lineage of Proboscidea (412.5 kg); and a 2.69× increase in body mass in the stem-lineage of Elephantimorpha (4114.39 kg), which is the last common ancestor of elephants and mastodons using the fossil record (*Figure 2A,B*). The ancestral Hyracoidea was inferred to be relatively small (2.86–118.18kg), and rate accelerations were coincident with independent body mass increases in large hyraxes such as *Titanohyrax andrewsi* (429.34 kg, 67.36× increase) (*Figure 2A,B*). While the body mass of the ancestral Sirenian was inferred to be large (61.7–955.51 kg), a rate acceleration occurred coincident with a 10.59× increase in body mass in Stellar's sea cow (*Figure 2A,B*). Rate accelerations also occurred coincident with dramatic reductions in body mass (36.6× decrease) in the stem-lineage of the dwarf elephants *Elephas (Palaeoloxodon) antiquus falconeri* and *Elephas cypriotes* (*Figure 2A,B*). These data indicate that gigantism in Afrotherians evolved step-wise, from small to

medium bodies in the Pseudoungulata stem-lineage, medium to large bodies in the Tehthytherian stem-lineage and extinct hyraxes, and from large to exceptionally large bodies independently in the Proboscidean stem-lineage and Stellar's sea cow (*Figure 2A,B*).

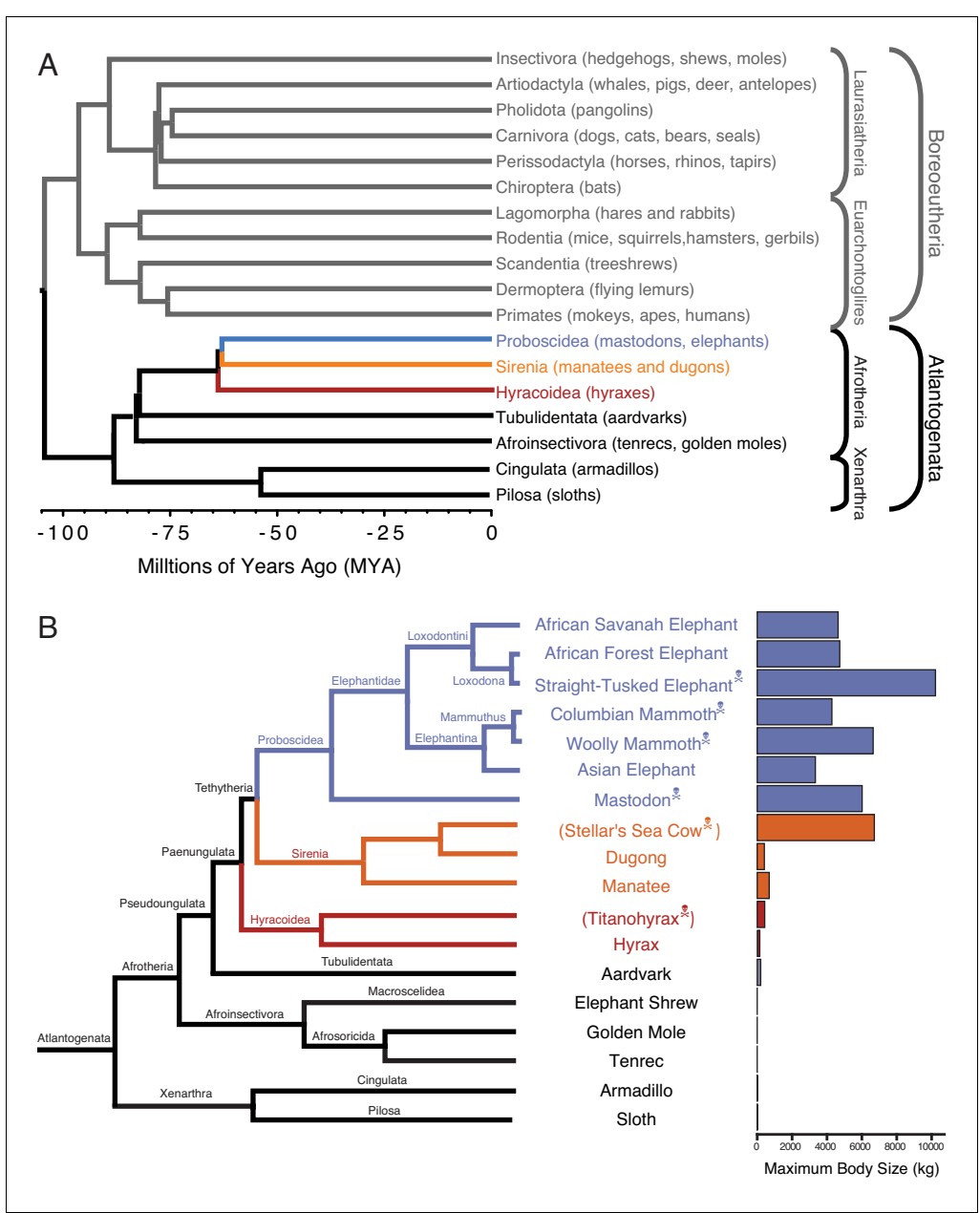

**Figure 1.** Large-bodied Afrotherians are nested within species with smaller body sizes (*Tacutu et al., 2013*; *Puttick and Thomas, 2015*). (**A**) Phylogenetic relationships between Eutherian orders, examples of each order are given in parenthesis. Horizontal branch lengths are proportional to time since divergence between lineages (see scale, Millions of Ago [MYA]). The clades Atlantogenata and Boreoeutheria are indicated, the order Proboscidea is colored blue, Sirenia is colored orange, and Hyracoidea is colored red. (**B**) Phylogenetic relationships of extant and recently extinct Atlantogenatans with available genomes are shown along with clade names and maximum body sizes. Note that horizontal branch lengths are arbitrary, species indicated with skull and crossbones are extinct, and those in parentheses do not have genomes. The order Proboscidea is colored blue, Sirenia is colored orange, and Hyracoidea is colored red.

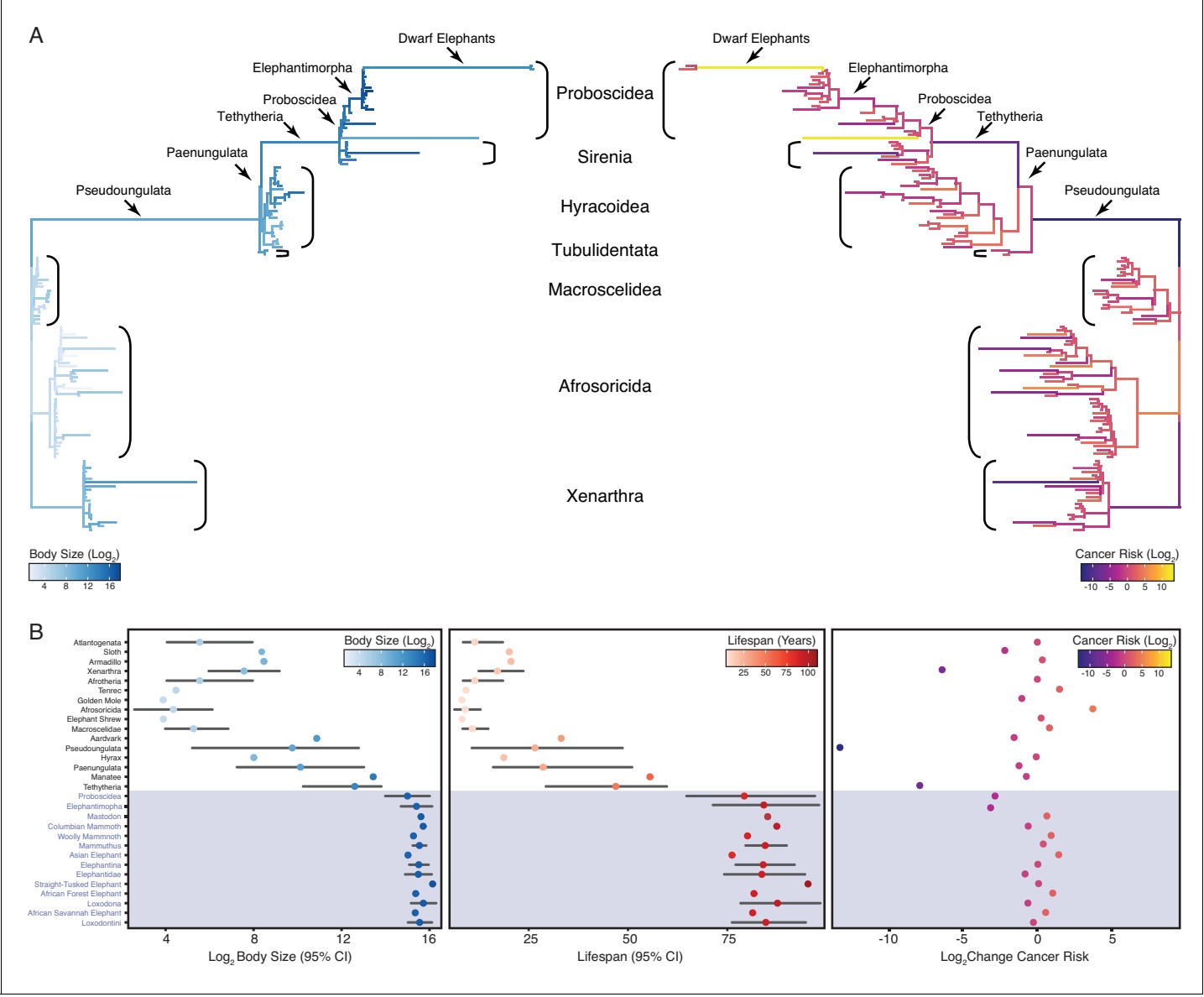

**Figure 2.** Convergent evolution of large-bodied, cancer resistant Afrotherians. (**A**) Atlantogenatan phylogeny, with branch lengths scaled by log$_2$ change in body size (left) or log$_2$ change in intrinsic cancer risk (right). Branches are colored according to ancestral state reconstruction of body mass or estimated intrinsic cancer risk. Clades and lineages leading to extant Proboscideans and dwarf elephants are labeled. (**B**) Extant and ancestral body size (left), lifespan (middle), and estimated intrinsic cancer risk reconstructions; data are shown as mean (dot) and 95% confidence interval (CI, whiskers).

## Step-wise reduction of intrinsic cancer risk in large, long-lived Afrotherians

In order to account for a relatively stable cancer rate across species (*Abegglen et al., 2015*; *Boddy et al., 2020*; *Tollis et al., 2020*), intrinsic cancer risk must also evolve with changes in body size and lifespan across species. We used empirical body size and lifespan data from extant species and empirical body size and estimated lifespan data from extinct species to estimate intrinsic cancer risk (*K*) with the simplified multistage cancer risk model $K \approx Dt^6$, where $D$ is the maximum body size and $t$ is the maximum lifespan (*Peto et al., 1975*; *Peto, 2015*; *Armitage, 1985*; *Armitage and Doll, 2004*). As expected, intrinsic cancer risk in Afrotheria also varies with changes in body size and longevity (*Figure 2A,B*), with a 6.41-log$_2$ decreases in the stem-lineage of Xenarthra, followed by a 13.37-log$_2$ decrease in Pseudoungulata, and a 1.49-log$_2$ decrease in Aardvarks (*Figure 2A*). In contrast to the Paenungulate stem-lineage, there is a 7.84-log$_2$ decrease in cancer risk in Tethytheria, a

0.67-log$_2$ decrease in Manatee, a 3.14-log$_2$ decrease in Elephantimorpha, and a 1.05-log$_2$ decrease in Proboscidea. Relatively minor decreases occurred within Proboscidea including a 0.83-log$_2$ decrease in Elephantidae and a 0.57-log$_2$ decrease in the American Mastodon. Within the Elephantidae, Elephantina and Loxodontini have a 0.06-log$_2$ decrease in cancer susceptibility, while susceptibility is relatively stable in Mammoths. The three extant Proboscideans, Asian Elephant, African Savana Elephant, and the African Forest Elephant, meanwhile, have similar decreases in body size, with slight increases in cancer susceptibility (*Figure 2A,B*).

## Pervasive duplication of tumor suppressor genes in Afrotheria

Our hypothesis was that genes which duplicated coincident with the evolution of increased body mass (IBM) and reduced intrinsic cancer risk (RICR) would be uniquely enriched in tumor suppressor pathways compared to genes that duplicated in other lineages. Therefore, we identified duplicated genes in each Afrotherian lineage (*Table 1* and *Figure 3A*) and tested if they were enriched in Reactome pathways related to cancer biology (*Figure 3B*, *Table 2*). No pathways related to cancer biology were enriched in either the Pseudoungulata (67.36-fold IBM, 13.37-log$_2$ RICR), but few genes were inferred to be duplicated in this lineage reducing power to detect enriched pathways. Consistent with our hypothesis, 18.18% of the pathways that were enriched in the Paenungulate stem-lineage (1.45-fold IBM, 1.17-log$_2$ RICR), 63% of the pathways that were enriched in the Tethytherian stem-lineage (11.82-fold IBM, 7.84-log$_2$ RICR), and 38.81% of the pathways that were enriched in the Proboscidean stem-lineage (1.06-fold IBM, 3.14-log$_2$ RICR) were related to tumor suppression (*Figure 3B*, *Table 2*). Similarly, 21.28% and 38.00% of the pathways that were enriched in manatee (1.11-fold IBM, 0.89-log$_2$ RICR) and aardvark (67.36-fold IBM, 1.49-log$_2$ RICR), respectively, were related to tumor suppression. In contrast, only 2.86% of the pathways that were enriched in hyrax (1.6-fold IBM, 1.49-log$_2$ RICR) were related to tumor suppression (*Figure 3B*, *Table 2*). Unexpectedly, however, lineages without major increases in body size or lifespan, or decreases in intrinsic cancer risk, were also enriched for tumor suppressor pathways. For example, 13.85%, 37.04%, and 22.00% of the pathways that were enriched in the stem-lineages of Afroinsectivoa and Afrosoricida, and in *E. telfairi*, respectively, were related to cancer biology (*Figure 3B*, *Table 2*).

Our observation that gene duplicates in most lineages are enriched in cancer pathways suggest either that duplication of genes in cancer pathways is common in Afrotherians, or that there may be a systemic bias in the pathway enrichment analyses. For example, random gene sets may be generally enriched in pathway terms related to cancer biology. To explore this latter possibility, we generated 5000 randomly sampled gene sets of between 10 and 5000 genes, and tested for enriched Reactome pathways using ORA. We found that no cancer pathways were enriched (median hypergeometric p-value ≤0.05) among gene sets tested greater than 157 genes; however, in these smaller gene sets, 12–18% of enriched pathways were classified as cancer pathways. Without considering p-value thresholds, the percentage of enriched cancer pathways approaches ~15% (213/1381) in simulated sets. Thus, for larger gene sets, we used a simulated threshold of ~15% to determine if pathways related to cancer biology were enriched more than one would expect from sampling bias (*Table 2*). We directly compared our simulated and observed enrichment results by lineage and gene set size, and found that Afrosoricida, Cape golden mole, tenrec, Elephantidae, elephant shrew, Asian elephant, African Savannah elephant, African Forest elephant, Columbian mammoth, aardvark, Paenungulata, Proboscidea, Tethytheria, and manatee had enriched cancer pathway percentages above background with respect to their gene set sizes, that is expected enrichments based on random sampling of small gene sets (*Table 2*). Thus, we conclude that duplication of genes in cancer pathways is common in many Afrotherians but that the inference of enriched cancer pathway duplication is not different from background in some lineages, particularly in ancestral nodes with a small number of estimated duplicates.

## Tumor suppressor pathways enriched exclusively within Proboscideans

While duplication of cancer associated genes is common in Afrotheria, the 157 genes that duplicated in the Proboscidean stem-lineage (*Figure 3A*) were uniquely enriched in 12 pathways related to cancer biology (*Figure 3B*). Among these uniquely enriched pathways (*Figure 3C*) were pathways related to the cell cycle, including 'G0 and Early G1', 'G2/M Checkpoints', and 'Phosphorylation of the APC/C', pathways related to DNA damage repair including 'Global Genome Nucleotide Excision

**Table 1.** Genomes used in this study.

| Species | Common Name | Genomes | Highest Quality Genome | Reference(s) |
|---|---|---|---|---|
| Choloepus hoffmanni | Hoffmans two-toed sloth | choHof1, choHof2, choHof-C_hoffmanni-2.0.1_HiC | choHof-C_hoffmanni-2.0.1_HiC | Dudchenko et al., 2017 |
| Chrysochloris asiatica | Cape golden mole | chrAsi1m | chrAsi1m | GCA_000296735.1 |
| Dasypus novemcinctus | Nine-banded armadillo | dasNov3 | dasNov3 | GCA_000208655.2 |
| Echinops telfairi | Lesser Hedgehog Tenrec | echTel2 | echTel2 | GCA_000313985.1 |
| Elephantulus edwardii | Cape elephant shrew | eleEdw1m | eleEdw1m | GCA_000299155.1 |
| Elephas maximus | Asian elephant | eleMaxD | eleMaxD | Palkopoulou et al., 2018 |
| Loxodonta africana | African savanna elephant | loxAfr3, loxAfrC, loxAfr4 | loxAfr4 | ftp://ftp.broadinstitute.org/pub/assemblies/mammals/elephant/loxAfr4 |
| Loxodonta cyclotis | African forest elephant | loxCycF | loxCycF | Palkopoulou et al., 2018 |
| Mammut americanum | American mastodon | mamAmel | mamAmel | Palkopoulou et al., 2018 |
| Mammuthus columbi | Columbian mammoth | mamColU | mamColU | Palkopoulou et al., 2018 |
| Mammuthus primigenius | Woolly mammoth | mamPriV | mamPriV | Palkopoulou et al., 2015 |
| Orycteropus afer | Aardvark | oryAfe1, oryAfe2 | oryAfe2 | Dudchenko et al., 2017 |
| Palaeoloxodon antiquus | Straight tusked elephant | palAntN | palAntN | Palkopoulou et al., 2018 |
| Procavia capensis | Rock hyrax | proCap1, proCap2, proCap-Pcap_2.0_HiC | proCap-Pcap_2.0_HiC | Dudchenko et al., 2017; Lindblad-Toh et al., 2011 |
| Trichechus manatus latirostris | Manatee | triMan1, triManLat2 | triManLat2 | Dudchenko et al., 2017; Foote et al., 2015 |

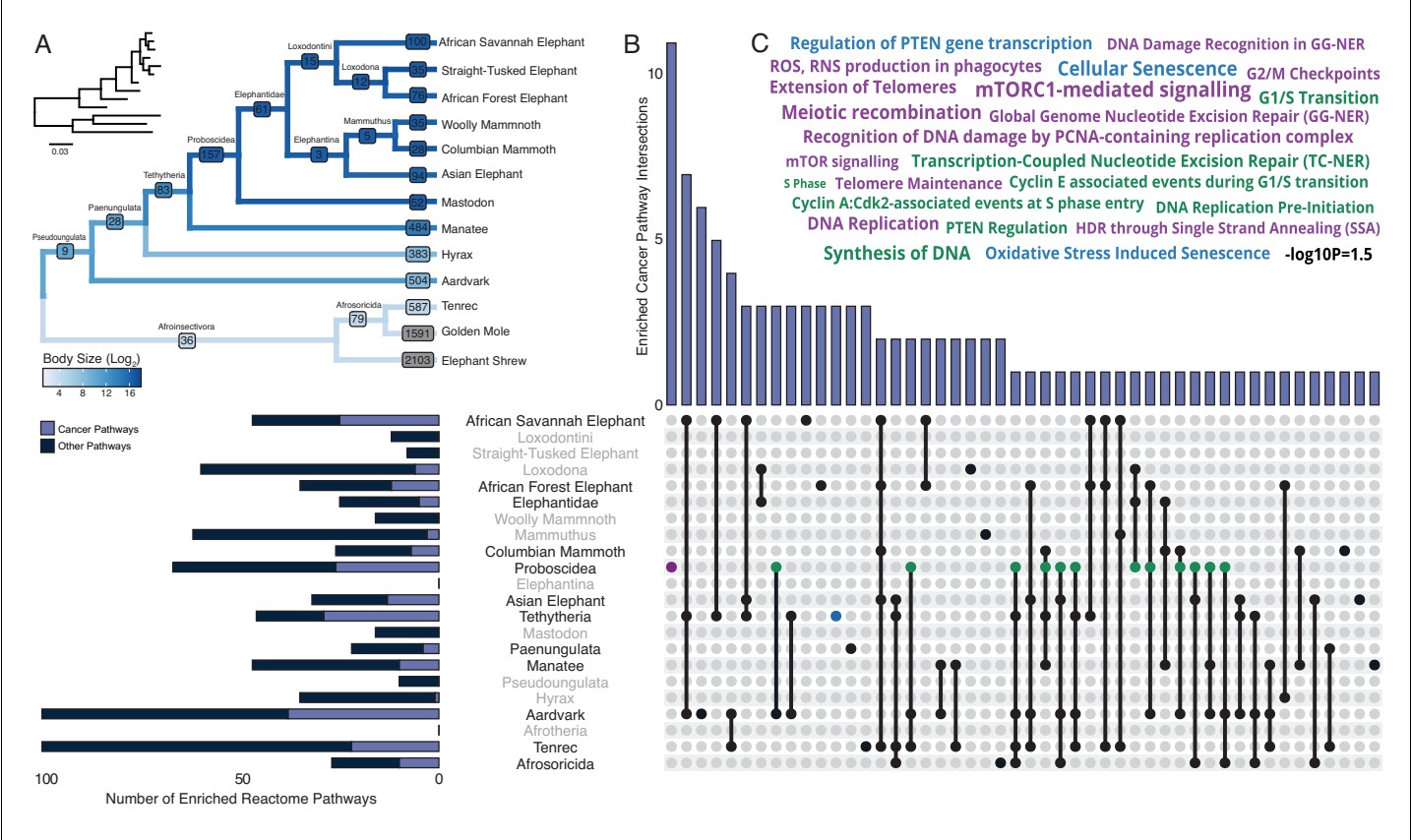

**Figure 3.** Pervasive duplication of tumor suppressors in Atlantogenata. (**A**) Afrotherian phylogeny indicating the number of genes duplicated in each lineage, inferred by maximum likelihood with Bayesian posterior probability (BPP) ≥0.80. Branches are colored according to log₂ change in body size. Inset, phylogeny with branch lengths proportional to gene expression changes per gene. (**B**) Upset plot of cancer related Reactome pathways enriched in each Afrotherian lineage; lineages in which the cancer pathway enrichment percentage is less than background are shown in gray. Note that Upset plots are Euler diagrams showing intersections between sets; lines indicate intersections in pathway terms between lineages connected by that line (for example, the line connecting the points for Aardvark and Tenrec indicate pathway indications for those two lineages), and empty sets are not shown. (**C**) Wordcloud of pathways enriched exclusively in the Proboscidean stem-lineage (purple), shared between Proboscidea and Tethytheria (blue), or shared between Proboscidea and any other lineage (green).

The online version of this article includes the following figure supplement(s) for figure 3:

**Figure supplement 1.** Estimated Copy Number by Coverage (ECNC) consolidates fragmented genes while accounting for missing domains in homologs.

**Figure supplement 2.** Correlations between genome quality metrics and ECNC metrics.

Repair (GG-NER)', 'HDR through Single Strand Annealing (SSA)', 'Gap-filling DNA repair synthesis and ligation in GG-NER', 'Recognition of DNA damage by PCNA-containing replication complex', and 'DNA Damage Recognition in GG-NER', pathways related to telomere biology including 'Extension of Telomeres' and 'Telomere Maintenance', pathways related to the apoptosome including 'Activation of caspases through apoptosome-mediated cleavage', and pathways related to 'mTORC1-mediated signaling' and 'mTOR signaling', which play important roles in the biology of aging. Thus, duplication of genes with tumor suppressor functions is pervasive in Afrotherians, but genes in some pathways related to cancer biology and tumor suppression are uniquely duplicated in large-bodied (long-lived) Proboscideans (*Figure 4A,B*).

Among the genes uniquely duplicated within Proboscideans are *TP53*, *COX20*, *LAMTOR5*, *PRDX1*, *STK11*, *BRD7*, *MAD2L1*, *BUB3*, *UBE2D1*, *SOD1*, *LIF*, *MAPRE1*, *CNOT11*, *CASP9*, *CD14*, and *HMGB2* (*Figure 4C*). Two of these, *TP53* and *LIF*, have been previously described (*Abegglen et al., 2015*; *Sulak et al., 2016*; *Vazquez et al., 2018*). These genes are significantly enriched in pathways involved in apoptosis, cell cycle regulation, and both upstream and downstream pathways involving

**Table 2.** Summary of reactome pathways in Atlantogenata.

| | Number of | | Percentage | | |
| --- | --- | --- | --- | --- | --- |
| | Genes | Pathways | Cancer pathways | Simulated cancer pathways | Cancer pathways greater than simulated? |
| *Afroinsectivora* | 36 | 65 | 13.85% | 15.42% | No |
| *Afrosoricida* | 79 | 27 | 37.04% | 15.42% | Yes |
| *Chrysochloris asiatica* | 1591 | 100 | 27.00% | 15.42% | Yes |
| *Echinops telfairi* | 587 | 100 | 22.00% | 15.42% | Yes |
| *Elephantidae* | 61 | 25 | 20.00% | 13.03% | Yes |
| *Elephantulus edwardii* | 2103 | 100 | 22.00% | 15.42% | Yes |
| *Elephas maximus* | 94 | 32 | 40.63% | 17.73% | Yes |
| *Loxodona* | 12 | 60 | 10.00% | 14.53% | No |
| *Loxodonta africana* | 100 | 47 | 53.19% | 15.42% | Yes |
| *Loxodonta cyclotis* | 76 | 35 | 34.29% | 16.11% | Yes |
| *Loxodontini* | 15 | 12 | 0.00% | 13.82% | No |
| *Mammut americanum* | 52 | 16 | 0.00% | 12.91% | No |
| *Mammuthus* | 5 | 62 | 4.84% | 15.29% | No |
| *Mammuthus columbi* | 28 | 26 | 26.92% | 12.88% | Yes |
| *Mammuthus primigenius* | 35 | 16 | 0.00% | 12.28% | No |
| *Orycteropus afer* | 504 | 100 | 38.00% | 15.42% | Yes |
| *Paenungulata* | 28 | 22 | 18.18% | 12.88% | Yes |
| *Palaeoloxodon antiquus* | 35 | 8 | 0.00% | 12.28% | No |
| *Proboscidea* | 157 | 67 | 38.81% | 9.52% | Yes |
| *Procavia capensis* | 383 | 35 | 2.86% | 15.42% | No |
| *Pseudoungulata* | 9 | 10 | 0.00% | 14.90% | No |
| *Tethytheria* | 83 | 46 | 63.04% | 18.52% | Yes |
| *Trichechus manatus* | 484 | 47 | 21.28% | 15.42% | Yes |

TP53. The majority of these genes are expressed in African Elephant transcriptome data (*Figure 4D*), suggesting that they maintained functionality after duplication.

## Coordinated duplication of TP53-related genes in Proboscidea

Prior studies found that the 'master' tumor suppressor *TP53* duplicated multiple times in elephants (*Abegglen et al., 2015*; *Sulak et al., 2016*), motivating us to further study duplication of genes involved in *TP53*-related pathways in Proboscidea. We traced the evolution of genes in the TP53 pathway that appeared in one or more Reactome pathway enrichments for genes duplicated recently in the African Elephant, which has the most complete genome among Proboscideans and for which several RNA-Seq data sets are available. We found that the initial duplication of TP53 in Tethytheria, where body size expanded, was preceded by the duplication of *GTF2F1* and *STK11* in Paenungulata and was coincident with the duplication of *BRD7*. These three genes are involved in regulating the transcription of *TP53* (*Liang and Mills, 2013*; *Launonen, 2005*; *Drost et al., 2010*; *Burrows et al., 2010*), and their duplication prior to that of *TP53* may have facilitated re-functionalization of *TP53* retroduplicates. Interestingly, *STK11* is also tumor suppressor that mediates tumor suppression via p21-induced senescence (*Launonen, 2005*). The other genes that are duplicated in the pathway are downstream of *TP53*; these genes duplicated either coincident with *TP53*, as in the case of *SIAH1*, or subsequently in Proboscidea, Elephantidae, or extant elephants (*Figure 4*). These genes are expressed in RNA-Seq data (*Figure 4D*), suggesting that they are functional.

While transcript abundance estimates inferred from RNA-Seq data can suggest that genes are functional, recent non-functional duplicates can still be transcribed. Therefore we inferred if each duplicate shown in *Figure 4C/D* encoded a putatively function protein by manually curation,

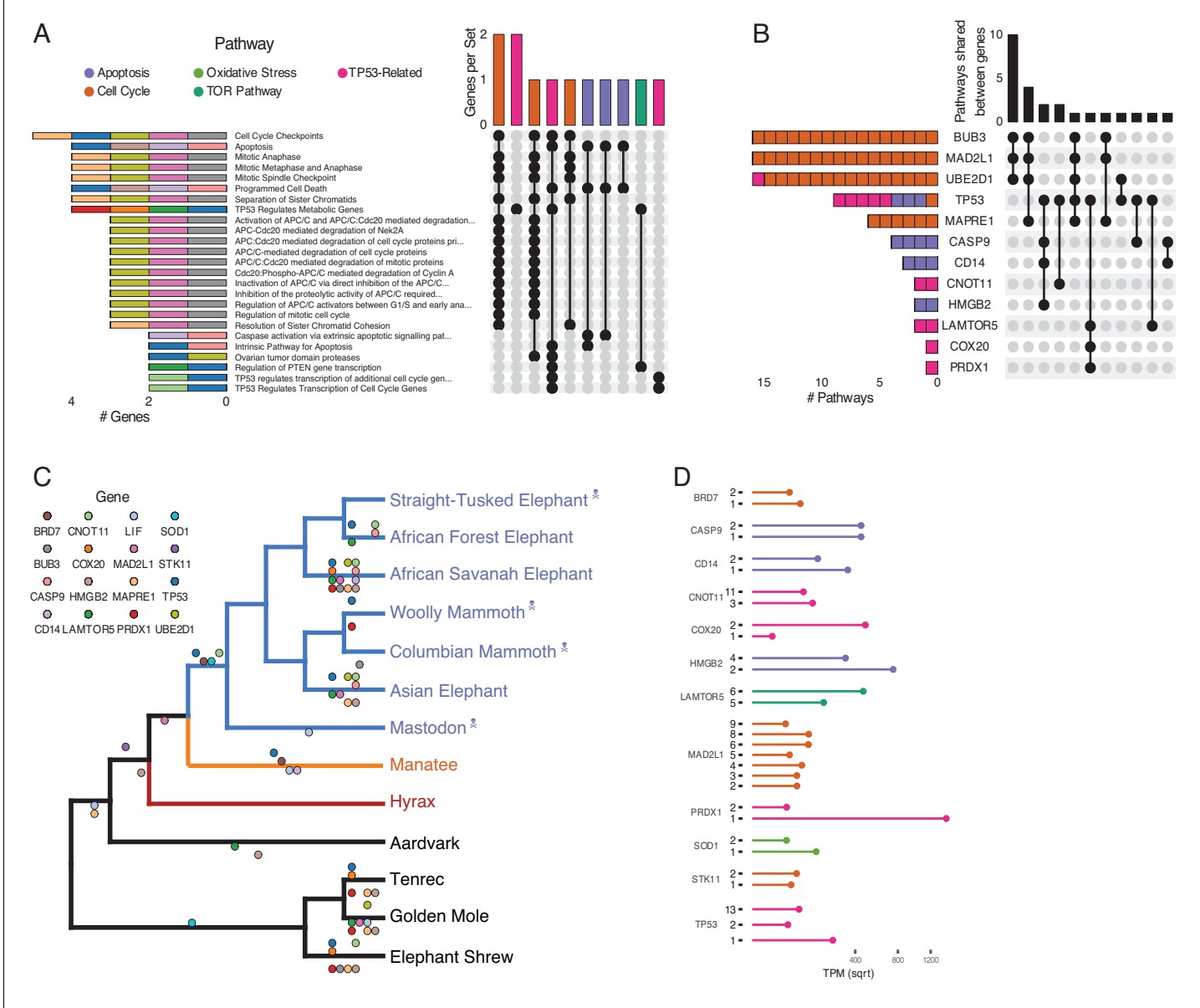

**Figure 4.** Duplications in the African savannah elephant (*Loxodonta africana*) are enriched for TP53-related and other tumor suppressor processes. (**A**) Upset plot of cancer-related Reactome pathways in African savannah elephant, highlighting shared genes in each set, and the pathway class represented by the combinations (see *Figure 3* for a description of Upset plots). (**B**) Inverted Upset plot from **A** showing the pathways shared by genes highlighted by WEBGESTALT in each pathway. (**C**) Cladogram of Afrotheria with sequenced genomes. Exemplar tumor suppressor duplicates are mapped onto lineages in which those genes are duplicated. Dots represent a duplication event of the color-coded genes. Note that we are unable to determine duplication status for some genes in Proboscideans because of assembly gaps in ancient genomes (indicated with skull and crossbones); these genes appear to be independently duplicated in extant species (African Forest, African Savanah, and Asian elephants) because they are missing from ancient genomes, biasing ancestral reconstructions of duplication status. (**D**) Gene expression levels of genes from panel **C** that have two or more expressed duplicates.

The online version of this article includes the following source data for figure 4:

**Source data 1.** Data set used for manual coding gene potential associated with *Figure 4C,D*.

specifically to identify premature stop codons and overall sequence conservation. Most genes in *Figure 4C/D*, such as *STK11*, *CD14*, *SOD1*, and *BRD7*, were well conserved and lacked premature stop codons. We also find that the *STK11*, *CD14*, and *BRD7* genes in the manatee were also well conserved, suggesting that extant manatees may also have enhanced tumor suppression and an

augmented stress response. However, some of the duplicate genes in the mantatee genome have premature stop codons suggesting they are not translated into functional proteins, including the additional copies of *MAPRE1*, *BUB3*, and *COX20* as well as at least one of the duplicate copies of *CNOT11*, *HMGB2*, *MAD2L1*, *LIF*, and *TP53*. For *TP53*, we have previously shown that duplicate copies of genes containing premature stop codons may still serve a functional role in regulating its progenitor's function. Thus, some of the genes with premature stop codons, such as duplicate *COX20* and *MAD2L1* which are expressed in RNA-Seq data, may encode functional lncRNA transcripts or truncated proteins. Some copies, including for *CASP9* and *PRDX1*, contained partial RBBH hits with no premature stop codons; however, they also lacked the totality of the coding sequence and thus may represent cases of pseudogenization, subfunctionalization, or neofunctionalization.

## Discussion

Among the evolutionary, developmental, and life history constraints on the evolution of large bodies and long lifespans is an increased risk of developing cancer. While body size and lifespan are correlated with cancer risk within species, there is no correlation between species because large and long-lived organisms have evolved enhanced cancer suppression mechanisms. While this ultimate evolutionary explanation is straightforward (*Peto, 2015*), determining the mechanisms that underlie the evolution of enhanced cancer protection is challenging because many mechanisms with relatively small effects likely contribute to evolution of reduced cancer risk. Previous candidate gene studies in elephants have identified duplications of tumor suppressors such as *TP53* and *LIF*, among others, suggesting that an increased copy number of tumor suppressors may contribute to the evolution of large body sizes in the elephant lineage (*Abegglen et al., 2015*; *Sulak et al., 2016*; *Vazquez et al., 2018*; *Caulin et al., 2015*; *Doherty and de Magalhães, 2016*). Here we: (1) trace the evolution of body size and lifespan in Eutherian mammals, with particular reference to Afrotherians; (2) infer changes in cancer susceptibility across Afrotherian lineages; (3) use a genome-wide screen to identify gene duplications in Afrotherian genomes, including multiple living and extinct Proboscideans; and (4) show that while duplication of genes with tumor suppressor functions is pervasive in Afrotherian genomes, Proboscidean gene duplicates are enriched in unique pathways with tumor suppressor functions.

### Correlated evolution of large bodies and reduced cancer risk

The hundred- to hundred-million-fold reductions in intrinsic cancer risk associated with the evolution of large body sizes in some Afrotherian lineages, in particular Elephantimorphs such as elephants and mastodons, suggests that these lineages must have also evolved remarkable mechanisms to suppress cancer. While our initial hypothesis was that large-bodied lineages would be uniquely enriched in duplicate tumor suppressor genes compared to other smaller-bodied lineages, we unexpectedly found that the duplication of genes in tumor suppressor pathways occurred at various points throughout the evolution of Afrotheria, regardless of body size. These data suggest that this abundance of tumor suppressors may have contributed to the evolution of large bodies and reduced cancer risk, but that these processes were not necessarily coincident. Interestingly, pervasive duplication of tumor suppressors may also have contributed to the repeated evolution of large bodies in hyraxes and sea cows, because at least some of the genetic changes that underlie the evolution of reduced cancer risk were common in this group. It remains to be determined whether our observation of pervasive duplication of tumor suppressors also occurs in other multicellular lineages. Using a similar reciprocal best BLAST/BLAT approach that focused on estimating copy number of known tumor suppressors in mammalian genomes, for example, *Caulin et al., 2015* found no correlation between copy number or tumor suppressors with either body mass or longevity, whereas *Tollis et al., 2020* found a correlation between copy number and longevity (but not body size) (*Tollis et al., 2020*; *Caulin et al., 2015*). These opposing conclusions may result from differences in the number of genes (81 vs 548) and genomes (8 vs 63) analyzed, highlighting the need for genome-wide analyses of many species that vary in body size and longevity.

## There's no such thing as a free lunch: Trade-offs and constraints on tumor suppressor copy number

While we observed that duplication of genes in cancer related pathways – including genes with known tumor suppressor functions – is pervasive in Afrotheria, the number of duplicate tumor suppressor genes was relatively small, which may reflect a trade-off between the protective effects of increased tumor suppressor number on cancer risk and potentially deleterious consequences of increased tumor suppressor copy number. Overexpression of *TP53* in mice, for example, is protective against cancer but associated with progeria, premature reproductive senescence, and early death; however, transgenic mice with a duplication of the *TP53* locus that includes native regulatory elements are healthy and experience normal aging, while also demonstrating an enhanced response to cellular stress and lower rates of cancer (*García-Cao et al., 2002*; *Tyner et al., 2002*). These data suggest that duplication of tumor suppressors can contribute to augmented cancer resistance, if the duplication includes sufficient regulatory architecture to direct spatially and temporally appropriate gene expression. Thus, it is interesting that duplication of genes that regulate TP53 function, such as *STK11*, *SIAH1*, and *BRD7*, preceded the retroduplication *TP53* in the Proboscidean stem-lineage, which may have mitigated toxicity arising from dosage imbalances. Similar co-duplication events may have alleviated the negative pleiotropy of tumor suppressor gene duplications to enable their persistence and allow for subsequent co-option during the evolution of cancer resistance.

## Caveats and limitations

Our genome-wide results suggest that duplication of tumor suppressors is pervasive in Afrotherians and may have enabled the evolution of larger body sizes in multiple lineages by lowering intrinsic cancer risk either prior to or coincident with increasing body size. However, our study has several inherent limitations. For example, we have shown that genome quality plays an important role in our ability to identify duplicate genes, and several species have poor quality genomes (and thus were excluded from further analyses). While several efforts have been established with the goal of generating high quality (chromosome length) reference genomes for mammals, such as DNAZoo, The Zoonomia Project, the Vertebrate Genomes Project, and Genome 10K, Atlantogenatans represent a minority of available genome projects. And while a few high quality Atlantogenatan genomes are available, they lack reference gene and transcriptome annotations, and genome browser graphical user interfaces that allow for easy access to genome data for the broader community, limiting their usefulness. Similarly, without comprehensive gene expression data we cannot be certain that duplicate genes are actually expressed, and thus functional. Our results on genome quality suggest several research priorities for these less well-studies species, including generating chromosome length reference genomes and genome annotations, and incorporating these species into existing genome browsers (such as UCSC Genome Browser).

We also assume that gene duplicates either maintain ancestral tumor suppressor functions and increase cancer resistance through dosage effects or provide redundancy to loss of function mutations thereby increasing robustness of tumor suppression. Many processes, such as developmental systems drift, neofunctionalization, and sub-functionalization, can cause divergence in gene functions and invalidate the assumption of conservation of gene function (*Rastogi and Liberles, 2005*; *Qian and Zhang, 2014*; *Stoltzfus, 1999*), leading to inaccurate inferences in gene and pathway functions which is a common problem in comparative genomic studies using pathway and gene ontologies to categorize gene function. In addition, we assume that most duplicate genes are functional but it is likely that some of the duplicates were identify are non-functional pseudogenes. Differentiating between functional and non-functional genes using comparative genomics can be challenging. For example, non-functional pseudogenes often accumulate non-synonymous amino acid substitutions and premature stop codons but these same changes can also occur in functional genes. For example, we have found that the elephant genome encodes *TP53* retogenes (*TP53RTGs*) all of which encode premature stop codons suggesting they are pseudogenes, but these *TP53TRG*s are expressed, encode functional separation of function mutants of the ancestral TP53 gene, and contribute to enhanced DNA damage sensitivity in elephant cells. Similarly, we have characterized duplicate *LIF* gene in elephants (*LIF6*) that lacks the start codon and exon 1 of the parent LIF gene. *LIF6* is expressed, encodes a functional protein with translation initiated at an alternative downstream start site, and also contributes to enhanced DNA damage sensitivity in elephant cells. In

addition, duplicate genes that lack coding potential, such as *PTENP1*, can also be expressed and while not translated function as LINC RNAs (in this case acting as a sponge for microRNAs that target the parent PTEN transcript). In each case classifying duplicates into putatively functional and non-functional categories based on sequence characteristic would misclassify *TP53RTG*s, *LIF6*, and *PTENP1*. Thus, sequence features of pseudogenes may maintain function, as a consequence of not excluding putative pseudogenes some of the genes we include in downstream analyses may be non-functional. Further experimental studies are needed to determine which duplicates are expressed and functional.

The focus of this study, motivated by our previous identification of *TP53* and *LIF* duplicates, was on the role gene duplication in general may have played in the resolution of Peto's paradox in large-bodied Afrotherians, particularly Proboscidea. Duplication of tumor suppressor genes, however, is unlikely to be the sole mechanism responsible for the evolution of large body sizes, long lifespans, and reduced cancer risk. The evolution of regulatory elements, coding genes, genes with non-canonical tumor suppressor functions, and immune cell recognition of cancerous cells are also likely important for reducing the risk of cancer.

## Conclusions: All Afrotherians are equal, but some are more equal than others

While we found that duplication of tumor suppressor genes is common in Afrotheria, genes that duplicated in the Proboscidean stem-lineage (*Figure 3A,B*) were uniquely enriched in functions and pathways that may be related to the evolution of unique anti-cancer cellular phenotypes in the elephant lineage (*Figure 3C*). Elephant cells, for example, cannot be experimentally immortalized (*Fukuda et al., 2016*; *Gomes et al., 2011*), rapidly repair DNA damage (*Sulak et al., 2016*; *Hart and Setlow, 1974*; *Francis et al., 1981*), are extremely resistant to oxidative stress (*Gomes et al., 2011*), and yet are also extremely sensitive to DNA damage (*Abegglen et al., 2015*; *Sulak et al., 2016*; *Vazquez et al., 2018*). Several pathways related to DNA damage repair, in particular nucleotide excision repair (NER), were uniquely enriched among genes that duplicated in the Proboscidean stem-lineage, suggesting a connection between duplication of genes involved in NER and rapid DNA damage repair (*Hart and Setlow, 1974*; *Francis et al., 1981*). Similarly, we identified a duplicate *SOD1* gene in Proboscideans that may confer the resistance of elephant cells to oxidative stress (*Gomes et al., 2011*). Pathways related to the cell cycle were also enriched among genes that duplicated in Proboscideans, and cell cycle dynamics are different in elephants compared to other species; population doubling (PD) times for African and Asian elephant cells are 13–16 days, while PD times are 21–28 days in other Afrotherians (*Gomes et al., 2011*). Finally, the role of 'mTOR signaling' in the biology of aging is well known. Collectively these data suggest that gene duplications in Proboscideans may underlie some of their cellular phenotypes that contribute to cancer resistance.

## Materials and methods

### Ancestral body size reconstruction

We first assembled a time-calibrated supertree of Eutherian mammals by combining the time-calibrated molecular phylogeny of *Bininda-Emonds et al., 2007*; *Bininda-Emonds et al., 2008* with the time-calibrated total evidence Afrotherian phylogeny from *Puttick and Thomas, 2015*. While the *Bininda-Emonds et al., 2007*; *Bininda-Emonds et al., 2008* phylogeny includes 1679 species, only 34 are Afrotherian, and no fossil data are included. The inclusion of fossil data from extinct species is essential to ensure that ancestral state reconstructions of body mass are not biased by only including extant species. This can lead to inaccurate reconstructions, for example, if lineages convergently evolved large body masses from a small-bodied ancestor. In contrast, the total evidence Afrotherian phylogeny of *Puttick and Thomas, 2015* includes 77 extant species and fossil data from 39 extinct species. Therefore, we replaced the Afrotherian clade in the *Bininda-Emonds et al., 2008* phylogeny with the Afrotherian phylogeny of *Puttick and Thomas, 2015* using Mesquite. Next, we jointly estimated rates of body mass evolution and reconstructed ancestral states using a generalization of the Brownian motion model that relaxes assumptions of neutrality and gradualism by considering increments to evolving characters to be drawn from a heavy-tailed stable distribution (the 'Stable Model')

implemented in StableTraits (*Elliot and Mooers, 2014*). The stable model allows for large jumps in traits and has previously been shown to outperform other models of body mass evolution, including standard Brownian motion models, Ornstein–Uhlenbeck models, early burst maximum likelihood models, and heterogeneous multi-rate models (*Elliot and Mooers, 2014*).

## Reciprocal Best Hit BLAT

We developed a reciprocal best hit BLAT (RBHB) pipeline to identify putative homologs and estimate gene copy number across species. The Reciprocal Best Hit (RBH) search strategy is conceptually straightforward: (1) Given a gene of interest $G_A$ in a query genome $A$, one searches a target genome $B$ for all possible matches to $G_A$; (2) For each of these hits, one then performs the reciprocal search in the original query genome to identify the highest-scoring hit; (3) A hit in genome $B$ is defined as a homolog of gene $G_A$ if and only if the original gene $G_A$ is the top reciprocal search hit in genome $A$. We selected BLAT (*Kent, 2002*) as our algorithm of choice, as this algorithm is sensitive to highly similar (>90% identity) sequences, thus identifying the highest-confidence homologs while minimizing many-to-one mapping problems when searching for multiple genes. RBH performs similar to other more complex methods of orthology prediction and is particularly good at identifying incomplete genes that may be fragmented in low quality/poorly assembled regions of the genome (*Altenhoff and Dessimoz, 2009*; *Salichos and Rokas, 2011*).

## Effective copy number by coverage

In low-quality genomes, many genes are fragmented across multiple scaffolds, which results in BLA (S)T-like methods calling multiple hits when in reality there is only one gene. To compensate for this, we developed a novel statistic, Estimated Copy Number by Coverage (ECNC), which averages the number of times we hit each nucleotide of a query sequence in a target genome over the total number of nucleotides of the query sequence found overall in each target genome (*Figure 3—figure supplement 1*). This allows us to correct for genes that have been fragmented across incomplete genomes, while accounting for missing sequences from the human query in the target genome. Mathematically, this can be written as:

$$\mathrm{ECNC} = \frac{\sum_{n=1}^{l} C_n}{\sum_{n=1}^{l} \mathrm{bool}(C_n)} \tag{1}$$

where $n$ is the given nucleotide in the query, $l$ is the total length of the query, $C_n$ is the number of instances that $n$ is present within a reciprocal best hit, and bool $(C_n)$ is 1 if $C_n$ >1 $C_n$>0 or 0 if $C_n$ =1 $C_n = 0$.

## RecSearch pipeline

We created a custom Python pipeline for automating RBHB searches between a single reference genome and multiple target genomes using a list of query sequences from the reference genome. For the query sequences in our search, we used the hg38 UniProt proteome (*The UniProt Consortium, 2017*), which is a comprehensive set of protein sequences curated from a combination of predicted and validated protein sequences generated by the UniProt Consortium. Next, we excluded genes from downstream analyses for which assignment of homology was uncertain, including uncharacterized ORFs (991 genes), LOC (63 genes), HLA genes (402 genes), replication dependent histones (72 genes), odorant receptors (499 genes), ribosomal proteins (410 genes), zinc finger transcription factors (1983 genes), viral and repetitive-element-associated proteins (82 genes), and 'Uncharacterized', 'Putative', or 'Fragment' proteins (30,724 genes), leaving a final set of 37,582 query protein isoforms, corresponding to 18,011 genes. We then searched for all copies of 18,011 query genes in publicly available Afrotherian genomes (*Dobson, 2013*), including African savannah elephant (*Loxodonta africana*: loxAfr3, loxAfr4, loxAfrC), African forest elephant (*Loxodonta cyclotis*: loxCycF), Asian Elephant (*Elephas maximus*: eleMaxD), Woolly Mammoth (*Mammuthus primigenius*: mamPriV), Colombian mammoth (*Mammuthus columbi*: mamColU), American mastodon (*Mammut americanum*: mamAmel), Rock Hyrax (*Procavia capensis*: proCap1, proCap2, proCap2*HiC*), West Indian Manatee (*Trichechus manatus latirostris*: triManLat1, triManLat1HiC), Aardvark (*Orycteropus afer*: oryAfe1, oryAfe1*HiC*), Lesser Hedgehog Tenrec (*Echinops telfairi*: echTel2), Nine-banded armadillo (*Dasypus novemcinctus*: dasNov3), Hoffman's two-toed sloth (*Choloepus hoffmannii*: choHof1,

choHof2, choHof2HiC), Cape golden mole (*Chrysochloris asiatica*: chrAsi1), and Cape elephant shrew (*Elephantulus edwardii*: eleEdw1) (*Dudchenko et al., 2017*; *Palkopoulou et al., 2015*; *Palkopoulou et al., 2018*; *Foote et al., 2015*).

A summary of gene duplications in each species is available in *Supplementary file 1*.

## Duplication gene inclusion criteria

In order to condense transcript-level hits into single gene loci, and to resolve many-to-one genome mappings, we removed exons where transcripts from different genes overlapped, and merged overlapping transcripts of the same gene into a single gene locus call. The resulting gene-level copy number table was then combined with the maximum ECNC values observed for each gene in order to call gene duplications. We called a gene duplicated if its copy number was two or more, and if the maximum ECNC value of all the gene transcripts searched was 1.5 or greater; previous studies have shown that incomplete duplications can encode functional genes (*Sulak et al., 2016*; *Vazquez et al., 2018*), therefore partial gene duplications were included provided they passed additional inclusion criteria (see below). The ECNC cut-off of 1.5 was selected empirically, as this value minimized the number of false positives seen in a test set of genes and genomes. The results of our initial search are summarized in *Figure 3A*. Overall, we identified 13,880 genes across all species, or 77.1% of our starting query genes.

## Genome quality assessment using CEGMA

In order to determine the effect of genome quality on our results, we used the gVolante webserver and CEGMA to assess the quality and completeness of the genome (*Nishimura et al., 2017*; *Parra et al., 2009*). CEGMA was run using the default settings for mammals ('Cut-off length for sequence statistics and composition'=1; 'CEGMA max intron length'=100,000; 'CEGMA gene flanks'=10,000, 'Selected reference gene set' = CVG). For each genome, we generated a correlation matrix using the aforementioned genome quality scores, and either the mean copy number or mean ECNC for all hits in the genome. We observed that the percentage of duplicated genes in non-Pseudoungulatan genomes was higher (12.94–23.66%) than Pseudoungulatan genomes (3.26–7.80%). Mean copy number, mean ECNC, and mean CN (the lesser of copy number and ECNC per gene) moderately or strongly correlated with genomic quality, such as LD50, the number of scaffolds, and contigs with a length above either 100K or 1M (*Figure 3—figure supplement 2*). The Afrosoricidians had the greatest correlation between poor genome quality and high gene duplication rates, including larger numbers of private duplications. The correlations between genome quality metric and number of gene duplications were particularly high for Cape golden mole (*Chrysochloris asiatica*: chrAsi1) and Cape elephant shrew (*Elephantulus edwardii*: eleEdw1); therefore we excluded these species from downstream pathway enrichment analyses.

## Determining functionality of duplicated via gene expression

In order to ascertain the functional status of duplicated genes, we generated de novo transcriptomes using publicly available RNA-sequencing data for African savanna elephant, West Indian manatee, and nine-banded armadillo (*Supplementary file 2*). We mapped reads to the highest quality genome available for each species, and assembled transcripts using HISAT2 and StringTie (*Kim et al., 2015*; *Pertea et al., 2015*; *Pertea et al., 2016*). We found that many of our identified duplicates had transcripts mapping to them above a Transcripts Per Million (TPM) score of 2, suggesting that many of these duplications are functional. RNA-sequencing data was not available for Cape golden mole, Cape elephant shrew, rock hyrax, aardvark, or the lesser hedgehog tenrec.

## Reconstruction of ancestral copy numbers

We encoded the copy number of each gene for each species as a discrete trait ranging from 0 (one gene copy) to 31 (for 32+ gene copies) and used IQ-TREE to select the best-fitting model of character evolution (*Minh et al., 2020*; *Hoang et al., 2018*; *Kalyaanamoorthy et al., 2017*; *Wang et al., 2018*; *Schrempf et al., 2019*), which was inferred to be a Jukes-Cantor type model for morphological data (MK) with equal character state frequencies (FQ) and rate heterogeneity across sites approximated by including a class of invariable sites (I) plus a discrete Gamma model with four rate categories (G4). Next we inferred gene duplication and loss events with the empirical Bayesian

ancestral state reconstruction (ASR) method implemented in IQ-TREE (*Minh et al., 2020*; *Hoang et al., 2018*; *Kalyaanamoorthy et al., 2017*; *Wang et al., 2018*; *Schrempf et al., 2019*), the best fitting model of character evolution (MK+FQ+GR+I) (*Soubrier et al., 2012*; *Yang et al., 1995*), and the unrooted species tree for Atlantogenata. We considered ancestral state reconstructions to be reliable if they had Bayesian Posterior Probability (BPP) $\geq 0.80$; less reliable reconstructions were excluded from pathway analyses. We note that there may be 'ghost' duplication events, that is genes that duplicated in, for example, the Tethytherian stem-lineage that are maintained in the Stellar's sea cow genome and lost in the manatee genome. These genes will be reconstructed as a Proboscidean-specific duplication events because we cannot determine copy number in extinct species that lack genomes.

## Pathway enrichment analysis

To determine if gene duplications were enriched in particular biological pathways, we used the WEB-based Gene SeT AnaLysis Toolkit (WebGestalt) (*Liao et al., 2019*) to perform Over-Representation Analysis (ORA) using the Reactome database (*Jassal et al., 2020*). Gene duplicates in each lineage were used as the foreground gene set, and the initial query set was used as the background gene set. WebGestalt uses a hypergeometric test for statistical significance of pathway over-representation, which we refined using two methods: a False Discovery Rate (FDR)-based approach and an empirical p-value approach (*Chen et al., 2013*). The Benjamini–Hochberg FDR multiple-testing correction was generated by WebGestalt. In order to correct p-values based on an empirical distribution, we modified the approach used by Chen et al. in Enrichr (*Chen et al., 2013*) to generate a 'combined score' for each pathway based on the hypergeometric p-value from WebGestalt, and a correction for expected rank for each pathway. In order to generate the table of expected ranks and variances for this approach, we randomly sampled foreground sets of 10–5000 genes from our background set 5000 times, and used WebGestalt ORA to obtain a list of enriched terms and P-values for each run; we then compiled a table of Reactome terms with their expected frequencies and standard deviation. These data were used to calculate a Z-score for terms in an ORA run, and the combined score was calculated using the formula $C = log(p) \cdot z$.

## Estimating the evolution of cancer risk

The dramatic increase in body mass and lifespan in some *Afrotherian* lineages, and the relatively constant rate of cancer across species of diverse body sizes (*Abegglen et al., 2015*), indicates that those lineages must have also evolved reduced cancer risk. To infer the magnitude of these reductions we estimated differences in intrinsic cancer risk across extant and ancestral Afrotherians. Following *Peto, 2015*, we estimate the intrinsic cancer risk ($K$) as the product of risk associated with body mass and lifespan. In order to determine ($K$) across species and at ancestral nodes (see below), we first estimated ancestral lifespans at each node. We used Phylogenetic Generalized Least-Square Regression (PGLS) (*Felsenstein, 1985*; *Martins and Hansen, 1997*), using a Brownian covariance matrix as implemented in the R package *ape* (*Paradis and Schliep, 2019*), to calculate estimated ancestral lifespans across Atlantogenata using our estimates for body size at each node. In order to estimate the intrinsic cancer risk of a species, we first inferred lifespans at ancestral nodes using PGLS (*Supplementary file 3*) and the model. Next, we calculated $K_1$ at all nodes, and then estimated the fold-change in cancer susceptibility between ancestral and descendant nodes (*Figure 2*). Next, in order to calculate $K_1$ at all nodes, we used a simplified multistage cancer risk model for body size $D$ and lifespan $t$: $K \approx Dt^6$ (*Peto et al., 1975*: *Peto, 2015*; *Armitage, 1985*; *Armitage and Doll, 2004*). The fold change in cancer risk between a node and its ancestor was then defined as $\log_2\left(\frac{K_2}{K_1}\right)$.

## Data analysis

All data analysis was performed using Python version 3.8 and R version 4.0.2 (2020-06-22), and the complete reproducible manuscript, along with code and data generation pipeline, can be found on our GitHub page at https://github.com/docmanny/atlantogenataGeneDuplication (*Vazquez and Lynch, 2021*; copy archived at swh:1:rev:6bc68ac31ef148131480710e50b0b75d06077db2; *Paradis and Schliep, 2019*; *Paradis et al., 2020*; *R Development Core Team, 2019*; *Xie, 2020*; *Bolker and Robinson, 2020*; *Dowle and Srinivasan, 2019*; *Wickham et al., 2020a*; *Wickham, 2020*;

*Harmon et al., 2020*; *Yutani, 2020*; *Yu, 2020a*; *Campitelli, 2020*; *Wickham et al., 2020b*; *Yu, 2020b*; *Kassambara, 2020*; *Slowikowski, 2020*; *Xiao, 2018*; *Yu and Lam, 2020c*; *Zhu, 2019*; *Ooms, 2020*; *Bache and Wickham, 2014*; *Pinheiro and Bates, 2020*; *Sievert et al., 2020*; *Henry and Wickham, 2020*; *Wickham et al., 2018*; *Hlavac, 2018*; *Wickham, 2019a*; *Müller and Wickham, 2020*; *Wickham and Henry, 2020*; *Yu, 2020d*; *Wickham, 2019b*; *Yu, 2020e*; *Gehlenborg, 2019*; *Xie, 2016*; *Alfaro et al., 2009*; *Eastman et al., 2011*; *Slater et al., 2012*; *Harmon et al., 2008*; *Pennell et al., 2014*; *Wickham, 2016*; *Yu, 2020f*; *Yu et al., 2018*; *Yu et al., 2017*; *Sievert, 2020*; *Wickham et al., 2019*; *Wang et al., 2020*). All files necessary to reproduce the data in this manuscript are provided in *Source data 1*.

## Manual verification of duplicate genes

We manually verified the coding potential of the 16 genes shown in *Figure 4* by first identifying the reciprocal best (DNA sequence) BLAT hits in the elephant and manatee genomes, which allowed us to determine conservation and presence of premature stop codons in the each open reading frame (ORF). We translated the ORF for each hit into amino acid sequences and grouped up hits for each gene into one FASTA file along with the UniProt protein sequences for the human, dog, cat, and cow orthologs. Using a pipeline hosted at NGPhlyogeny.fr (*Lemoine et al., 2019*), the homologs were aligned using MAFFT *Katoh and Standley, 2013*; the aligned sequences were cleaned using BMGE (*Criscuolo and Gribaldo, 2010*). Finally we used FastME (*Lefort et al., 2015*) to infer a gene tree for each duplicate. Alignments were then visually inspected for conservation and presence of premature stop codons.

# Acknowledgements

We would like to thank Dr. Olga Dudchenko and Dr. Erez Aiden at Baylor College of Medicine for the Hi-C scaffolded *Procavia capensis*, *Trichechus manatus*, *Orycteropus afer*, and *Choloepus hoffmannii* genomes. We would also like to thank D.H. Vazquez for his indispensable support.

# Additional information

## Funding

| Funder | Author |
| --- | --- |
| University of Chicago | Juan M Vazquez<br>Vincent J Lynch |

The funders had no role in study design, data collection and interpretation, or the decision to submit the work for publication.

## Author contributions

Juan M Vazquez, Conceptualization, Resources, Data curation, Software, Formal analysis, Validation, Investigation, Visualization, Methodology, Writing - original draft, Writing - review and editing; Vincent J Lynch, Conceptualization, Data curation, Formal analysis, Supervision, Investigation, Visualization, Writing - original draft, Project administration, Writing - review and editing

## Author ORCIDs

Juan M Vazquez https://orcid.org/0000-0001-8341-2390
Vincent J Lynch https://orcid.org/0000-0001-5311-3824

## Decision letter and Author response

Decision letter https://doi.org/10.7554/eLife.65041.sa1
Author response https://doi.org/10.7554/eLife.65041.sa2

# Additional files

## Supplementary files

- Source data 1. All necessary data sets and scripts to reproduce results presented in this manuscript.
- Supplementary file 1. Summary of duplications in Atlantogenata.
- Supplementary file 2. RNA-Seq data sets used in this study, along with key biological and genome information.
- Supplementary file 3. Summary of PGLS model used to estimate lifespan.
- Transparent reporting form

## Data availability

All data generated or analysed during this study are included in the manuscript and supporting files.

The following previously published datasets were used:

| Author(s) | Year | Dataset title | Dataset URL | Database and Identifier |
|---|---|---|---|---|
| Di Palma F, Alfoldi J, Johnson J, Berlin A, Gnerre S, Jaffe D, MacCallum I, Young S, Walker BJ, Lindblad-Toh K | 2013 | Chrysochloris asiatica (Cape golden mole) genome | https://www.ncbi.nlm.nih.gov/assembly/GCF_000296735.1/ | NCBI Assembly, ChrAsi1.0 |
| Gnerre S, Heiman D, Young S, Fulton L, Delehaunty K, Minx P, Chinwalla A, Mardis E, Wilson R, Warren W | 2018 | The Genome Sequence of Choloepus hoffmanni (sloth) | https://www.dnazoo.org/assemblies/Choloepus_hoffmanni | NCBI Assembly, C_hoffmanni-2.0.1_HiC |
| Lindblad-Toh K, Chang JL, Gnerre S, Clamp M, Lander ES | 2012 | Dasypus novemcinctus (nine-banded armadillo) genome | https://www.ncbi.nlm.nih.gov/assembly/GCA_000208655.2/ | NCBI Assembly, Dasnov3.0 |
| Di Palma F, Alfoldi J, Johnson J, Berlin A, Gnerre S, Jaffe D, MacCallum I, Young S, Walker BJ, Lindblad-Toh K | 2013 | Echinops telfairi (small Madagascar hedgehog) | https://www.ncbi.nlm.nih.gov/assembly/GCA_000313985.1/ | NCBI Assembly, EchTel2.0 |
| Di Palma F, Alfoldi J, Johnson J, Berlin A, Gnerre S, Jaffe D, MacCallum I, Young S, Walker BJ, Lindblad-Toh K | 2013 | Elephantulus edwardii (Cape elephant shrew) | https://www.ncbi.nlm.nih.gov/assembly/GCA_000299155.1/ | NCBI Assembly, EleEdw1.0 |
| Palkopoulou E, Lipson M, Mallick S, Nielsen S, Rohland N, Baleka S, Karpinski E, Ivancevic AM, To TH, Kortschak RD, Raison JM, Qu Z, Chin TJ, Alt KW, Claesson S, Dalén L, MacPhee RDE, Meller H, Roca AL, Ryder OA, Heiman D, Young S, Breen M, Williams C, Aken BL, Ruffier M, Karlsson E, Johnson J, Di Palma F, | 2018 | A comprehensive genomic history of extinct and living elephants | https://www.ebi.ac.uk/ena/browser/view/PRJEB24361 | NCBI Assembly, PRJEB24361 |

| Alfoldi J, Adelson DL, Mailund T, Munch K, Lindblad-Toh K, Hofreiter M, Poinar H, Reich D | | | | |
|---|---|---|---|---|
| Di Palma F, Alfoldi J, Johnson J, Berlin A, Gnerre S, Jaffe D, MacCallum I, Young S, Walker BJ, Lindblad-Toh K | 2017 | Aardvark (Orycteropus afer) genome | https://www.dnazoo.org/assemblies/Orycteropus_afer | NCBI Assembly, oryAfe2 |
| Di Palma F, Alfoldi J, Johnson J, Berlin A, Gnerre S, Jaffe D, MacCallum I, Young S, Walker BJ, Lindblad-Toh K | 2013 | A comprehensive genomic history of extinct and living elephants | https://www.dnazoo.org/assemblies/Procavia_capensis | NCBI Assembly, proCap-Pcap_2.0_HiC |
| Andrew DF, Liu Y, Thomas GWC, Vinar T, Alfoldi J, Deng J, Dugan S | 2015 | West Indian manatee (Trichechus manatus) genome | https://www.dnazoo.org/assemblies/Trichechus_manatus | NCBI Assembly, triManLat2 |

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

# Appendix 1

Summary of duplicate gene annotations associated with *Figure 4C,D*.

BRD7:

Three manatee copies, but tml_BRD7_3 has PMSs. Overall high sequence similarity.

BUB3:

#manatee=#elephant.

PMS in loxafr #2–3, PMS in triman #2–3.

High AA conservation even in pseudogenes.

Casp9:

Two extra elephant copies are identical in AA sequence but hits only encompass a 47-AA hit that is highly conserved with matching domain in CASP9.

CD14:

#Manatee = #Elephant, all four copies are highly conserved.

CNOT11:

9/11 elephant copies do contain PMS; however, many of these are still mostly full length and highly conserved. The ones with an early PMS are less well conserved.

COX20:

Second elephant copy has two PMSs, but is still well conserved.

HMGB2:

2/4 elephant copies have PMSs, all copies highly conserved. Manatee 2/3 copies have PMS, lower conservation.

LAMTOR5:

Only one elephant copy has a PMS; three very highly conserved copies, others with conserved domains.

LIF:

Manatee's have 4 of 13 copies previously reported in elephants. See *Vazquez et al., 2018*.

TP53:

Elephant: 9/19 with PMSs. Low N' conservation, but high conservation on C' end. See *Sulak et al., 2016*.

STK11:

Very highly conserved, with divergence in the second copies in the elephant and manatee.

SOD1:

Highly conserved elephant copies. Manatee copies are only partial hits, with moderate conservation.

PRDX1:

Extremely strong conservation for main copies. Second elephant copy is a partial hit and has some divergence from the main elephant copy.

MAPRE1:

2/3 elephant copies have early PMSs, but with subsequent ATGs. Very high conservation in main and 1/2 duplicate elephant copies, only partial hits on third copy. Very high conversation between manatee copies.

MAD2L1:

7/9 elephant copies have PMSs. 2/6 manatee copies have PMSs. However, all copies are very highly conserved.

