## [Decision Letter]

**Acceptance summary:**

The study couples careful phylogenetic work with genomics to demonstrate that all Afrotherians are equal, but some (those with large body sizes) are more equal than others with respect to genetic mechanisms that reduce cancer risk. This is a significant advance in our understanding of the genetic evolution of cancer risk with body size, and in so doing it considerably lengthens the list of candidate genes that contribute to tumor suppression in Afrotherian mammals.

**Decision letter after peer review:**

Thank you for submitting your article "Pervasive duplication of tumor suppressors in Afrotherians during the evolution of large bodies and reduced cancer risk" for consideration by *eLife*. Your article has been reviewed by three peer reviewers, and the evaluation has been overseen by a Reviewing Editor and Patricia Wittkopp as the Senior Editor. The following individuals involved in review of your submission have agreed to reveal their identity: Stephen C Stearns (Reviewer #1); Vera Gorbunova (Reviewer #2). Please note the changes in our policy on revisions we have made in response to COVID-19 (https://elifesciences.org/articles/57162).

The reviewers have discussed the reviews with one another and the Reviewing Editor has drafted this decision to help you prepare a revised submission.

Summary:

This study addresses the question of whether duplication of tumor suppressor genes occurred coincidently with the enlarged body size of Afrotherian mammals. By reconstructing ancestral body sizes, cancer risks and gene duplication events across the Afrotherian phylogeny, the study shows that both increased body sizes and reduced cancer risks evolved gradually. Reactome pathway enrichment analysis for gene duplicates showed that gene duplicates in both lineages with or without major increases in body size / lifespan / decreases in cancer risk are enriched in many cancer related pathways. However, the authors found that 157 genes duplicated in Proboscidean stem-lineage, in which extremely large species evolved, were uniquely enriched in 12 cancer pathways. These genes might facilitate the further body enlargement and cancer resistance evolution in Proboscideans. Most interestingly, the authors found that several genes both upstream and downstream of a famous tumor suppressor TP53 have also been duplicated, either before or after initial TP53 duplication. These genes are involved in transcriptional regulation of TP53 and may have facilitated re-functionalization of TP53 retroduplicates. Overall, this is an important and interesting study that can help us understand the evolution of body size, lifespan and cancer risk in mammals more deeply.

Revisions:

1) In general, the evolutionary fate of gene duplication includes: 1) Conservation of gene function; 2) Neofunctionalization; 3) Pseudogenization; 4) Subfunctionalization (doi:10.1016/S01695347(03)00033-8). To execute the function of tumor suppression, as this study focused on, gene duplicates were supposed to be functionally conserved or subfunctionalized. Gene duplicates that have been neofunctionalized or pseudogenized will not be helpful (also mentioned by authors in the Caveats section). Therefore, it might be more convincing to investigate the functional status of each gene duplicate, especially those in Figure 4C/D. In many cases, however, a related function, rather than an entirely new function, evolves by neofunctionalization after gene duplication, and also that to check new functions for a batch of genes is not realistic, the authors could simply check the coding sequences to ensure these genes duplicates are not pseudogenes and are functional. This is necessary because in Figure 4D, many genes have only 2 copies expressed. If one of them is a young pseudogene, it could be stochastically expressed and will encode a dysfunctional protein.

2) In Results section 3, the cancer pathway frequency data of many nodes seem to not be consistent with data shown in Table 2. For example, "55.8% (29/52) of the pathways that were enriched in the Tethytherian stem-lineage…, 27.8% (20/72) of the pathways that were enriched in the Proboscidean stem-lineage…were related to tumor suppression", the cancer pathway percentages shown in Table 2 for these 2 nodes are 63.4% and 38.81%, respectively. While the frequency data in Table 2 are consistent with Supplementary file 3. It is possible that the frequency data shown in the main text are specific to pathways of tumor suppression, rather than cancer related pathways. If this is the case, more detailed data should be shown somewhere else.

3) The titles of Results section 3 and section 4 are highly similar and actually the data in section 4 seems to be used to further solidify the conclusion of section 3. Therefore, is it possible to merge them into one single section?

4) A major concern with the paper is that while the methods are very well-thought out, there are some unresolvable limitations to the data available. As it stands, the manuscript does not sufficiently guide the reader through how these issues might influence the findings. One example of this problem is in the estimation of cancer risk. The risk is estimated on the basis of body size and life span. However, that lifespan is itself phylogenetically estimated from body size at least for the non-extant species. It is not clear from the manuscript whether all lifespans are so estimated, or whether observations are used for the lifespan of the extant species. If the later, caution is indicated, because lifespan data are highly uneven and often given as observed maximal lifespans, which can be misleading if taken from, for instance, zoo specimens. In either case, the manuscript needs to more clearly emphasize that these are statistically-predicted risks, not measured risks.

5) On a broader note, the authors have done their best with a dataset that suffers from a couple of problems. First, all of the extant very large-bodied animals form a single clade, with the hyrax as the sole small-bodied member of that clade. And since the *titanohyrax* is extinct, among the extant organisms (an available large-bodies species with genomes) there is then a true large-bodied clade of the Sirenia and elephants and relatives. We understand that other evolutionary data make it clear that these represent two (three including *titanohyrax*) independent transitions to large-body sizes. But with only the modern or nearly modern genomes to work with, we are not sure that the duplication inference procedures and their coupling to the body size analysis statistically represent more than a single observation (e.g., a default of a single transition to large size along the tethytheria branch).

6) Similarly, the authors observe what appears to be a number of independent duplications of tumor suppressors in African and Asian elephants: duplications that are lacking in many of the ancient genomes considered. The authors used rigorous statistical methods to correct for the fragmented nature of these ancient genomes, but it is very hard not to wonder if some of the data in Figure 4 is really not an artifact of using ancient genomes, where detecting recent gene duplications may be very difficult (several of the Asian and African elephant duplications in Figure 4 appear to be of the same genes). If these events are truly independent and not genome assembly/annotation artifacts, there is then an alternative hypothesis to propose. Thus, are the authors suggesting that there is a rapid turnover in the duplication of tumor suppressors, such that all elephants have such duplicates, but the particular duplications have short life spans and differ from species to species?

7) It would be nice to see a few more comments on the manatee genome and why it does (or doesn't) show the expected patterns for the genome evolution in the face of the evolution of larger body sizes.

8) Figures 3 and 4 would benefit from greatly expanded captions: we do not fully understand what is being illustrated in, for instance, Figure 3B-why are certain dots connected with lines? Intersections between what in the y-axis label?

---

## [Author Response]

Revisions:1) In general, the evolutionary fate of gene duplication includes: 1) Conservation of gene function; 2) Neofunctionalization; 3) Pseudogenization; 4) Subfunctionalization (doi:10.1016/S01695347(03)00033-8). To execute the function of tumor suppression, as this study focused on, gene duplicates were supposed to be functionally conserved or subfunctionalized. Gene duplicates that have been neofunctionalized or pseudogenized will not be helpful (also mentioned by authors in the Caveats section). Therefore, it might be more convincing to investigate the functional status of each gene duplicate, especially those in Figure 4C/D. In many cases, however, a related function, rather than an entirely new function, evolves by neofunctionalization after gene duplication, and also that to check new functions for a batch of genes is not realistic, the authors could simply check the coding sequences to ensure these genes duplicates are not pseudogenes and are functional. This is necessary because in Figure 4D, many genes have only 2 copies expressed. If one of them is a young pseudogene, it could be stochastically expressed and will encode a dysfunctional protein.

We agree with the reviewers that this is a very important concern. We assume that gene duplicates either maintain ancestral tumor suppressor functions, and increase cancer resistance though dosage effects or by providing redundancy to loss of function mutations thereby increasing robustness of tumor suppression, or subfunctionalize expression domains.

However, we are wary of excluding genes from our analyses that have premature stop codons and other sequence features of pseudogenes because such genes may maintain tumor suppressor functions. For example, we have identified duplicate *TP53* retrogenes (*TP53TRG*s) in elephants, all of which encode premature stop codons suggesting they are pseudogenes. However, these *TP53TRG*s are expressed, encode functional separation of function mutants of the ancestral *TP53* gene, and contribute to enhanced DNA damage sensitivity in elephant cells.

Similarly, we have characterized duplicate *LIF* gene in elephants that lack the start codon of the parent LIF gene. This duplicate gene (*LIF6*) is also expressed, encodes a functional transcript with translation initiated at an alternative downstream start site, and contributes to enhanced DNA damage sensitivity in elephant cells.

In addition, duplicate genes that lack coding potential, such as *PTENP1*, can also be expressed and while not translated function as LINC RNAs (in this case acting as a sponge for microRNAs that target the parent *PTEN* transcript). In each case classifying duplicates into putatively functional and non-functional categories would misclassify *TP53RTG*, *LIF6*, and *PTENP1*.

Because we believe this reviewer concern is a serious one, we have expanded our Caveats and limitations section to include a discussion of this concern and our rationale for including all duplicate genes in our analyses. In this expanded Caveats and limitations section we include a discussion of the evolutionary fates of duplicate genes, including a discussion of genes that encode functional transcripts and proteins but appear to be non-functional based on sequence characteristics. We also discuss the limitations of our approach.

Furthermore, we manually verified the genes discussed in Figure 4C/D to identify premature stop codons and assess divergence between duplicate copies. We have added a new section in the Materials and methods titled “Manual verification of duplicate genes” describing our verification process, and an additional paragraph to Results corresponding to our findings. For the reviewers’ convenience, we include them here:

Manual verification of duplicate genes. For a subset of 16 genes (Figure 4), we pulled the DNA sequence matches from the reciprocal best BLAT hits in the elephant and manatee genomes to determine conservation and presence of premature stop codons in the open reading frame (ORF). We translated the ORF for each hit into amino acid sequences, and grouped up hits for each gene into one FASTA file along with the UniProt protein sequences for the human, dog, cat, and cow orthologs. Using a pipeline hosted at NGPhlyogeny.fr[@Lemoine_2019], the homologs were aligned using MAFFT[@Katoh_2013]; the aligned sequences were cleaned using BMGE[@Criscuolo_2010]; and then a gene tree was constructed using FastME[@Lefort_2015]. Alignments were then visually inspected for conservation and presence of premature stop codons.

Results of manual verification: For the genes shown in Figure 4C and D, we did additional manual curation to determine if any of them contained premature stop codons, as well as to inspect conservation. For most of these genes, such as STK11, CD14, SOD1, and BRD7, all copies are strongly conserved, and lack premature stop codons. We also find that for STK11, CD14, and BRD7, manatees also possess an additional copy with high conservation, suggesting that extant manatees may also have enhanced tumor suppression and stress response. Some of these hits, however, do have premature stop codons. These include the additional copies of MAPRE1, BUB3, and COX20; and some copies of CNOT11, HMGB2, MAD2L1, LIF, and TP53. For TP53, we have previously shown that duplicate copies of genes containing premature stop codons may still serve a functional role in regulating its progenitor's function. As such, and especially in the case of the copies of COX20 and MAD2L1 which are expressed, the presence of a premature stop codon relative to the progenitor gene may not indicate a true pseudogenization and loss of functionality. Some copies, including for CASP9 and PRDX1, contained partial hits with no neighboring or overlapping hits; while these contained no premature stops, they also do not likely contain the totality of the coding sequence of their hits, and thus may represent cases of sub- or neo-functionalization.

2) In Results section 3, the cancer pathway frequency data of many nodes seem to not be consistent with data shown in Table 2. For example, "55.8% (29/52) of the pathways that were enriched in the Tethytherian stem-lineage…, 27.8% (20/72) of the pathways that were enriched in the Proboscidean stem-lineage…were related to tumor suppression", the cancer pathway percentages shown in Table 2 for these 2 nodes are 63.4% and 38.81%, respectively. While the frequency data in Table 2 are consistent with Supplementary file 3. It is possible that the frequency data shown in the main text are specific to pathways of tumor suppression, rather than cancer related pathways. If this is the case, more detailed data should be shown somewhere else.

The data shown in Table 2 and Supplementary file 3 are correct, and the discrepancy was due to a typo during writing. We thank the reviewers for catching this, and we have corrected the text in the manuscript to reflect the data in Table 2.

3) The titles of Results section 3 and section 4 are highly similar and actually the data in section 4 seems to be used to further solidify the conclusion of section 3. Therefore, is it possible to merge them into one single section?

We agree these sections are very similar and have combined them, as suggested.

4) A major concern with the paper is that while the methods are very well-thought out, there are some unresolvable limitations to the data available. As it stands, the manuscript does not sufficiently guide the reader through how these issues might influence the findings. One example of this problem is in the estimation of cancer risk. The risk is estimated on the basis of body size and life span. However, that lifespan is itself phylogenetically estimated from body size at least for the non-extant species. It is not clear from the manuscript whether all lifespans are so estimated, or whether observations are used for the lifespan of the extant species. If the later, caution is indicated, because lifespan data are highly uneven and often given as observed maximal lifespans, which can be misleading if taken from, for instance, zoo specimens. In either case, the manuscript needs to more clearly emphasize that these are statistically-predicted risks, not measured risks.

We agree that this is an important concern and that estimation of lifespan and cancer risk has inherent limitations for extinct species, we apologize if we did not sufficiently guide the reader through our rationale and logic. We used empirical body size and lifespan for extant species, and estimates of lifespan based on body size for extinct species, and both of these estimates to statistically-predict cancer risks. We also note that, at least for elephants, captive individuals have shorter lifespans than wild individuals (Lahdenperä, L. et al. Differences in age-specific mortality between wild-caught and captive-born Asian elephants. Nature Communications (2018) DOI: 10.1038/s41467-018-05515-8; Clubb R. et al. Compromised survivorship in zoo elephants. Science (2008). DOI: 10.1126/science.1164298). Thus, it is not necessarily the case that data from zoo specimens represent maximum longevity.

We have updated the “Step-wise reduction of intrinsic cancer risk in large, long-lived Afrotherians” Results section to ensure that our methods and rationale are clear. Specifically, we have edited the first paragraph of this section to start with: “In order to account for a relatively stable cancer rate across species [*10*–*12*], intrinsic cancer risk must also evolve with changes body size and lifespan across species. We used empirical body size and lifespan data from extant species and empirical body size and estimated lifespan data from extinct species to estimate intrinsic cancer risk (*K*) with the simplified multistage cancer risk model K≈Dt6, where D is maximum body size and t is maximum lifespan [*9*, *54*, *58*, *59*].”

5) On a broader note, the authors have done their best with a dataset that suffers from a couple of problems. First, all of the extant very large-bodied animals form a single clade, with the hyrax as the sole small-bodied member of that clade. And since the titanohyrax is extinct, among the extant organisms (an available large-bodies species with genomes) there is then a true large-bodied clade of the Sirenia and elephants and relatives. We understand that other evolutionary data make it clear that these represent two (three including titanohyrax) independent transitions to large-body sizes. But with only the modern or nearly modern genomes to work with, we are not sure that the duplication inference procedures and their coupling to the body size analysis statistically represent more than a single observation (e.g., a default of a single transition to large size along the tethytheria branch).

Phylogenetic analyses utilizing both molecular and morphological data indicate that large bodies independently evolved in hyraxes, sea cows, and elephants (Paenungulata). Unfortunately, without genomes from the largest hyrax and sirenian species (such as *Titanohyrax* and Stellar’s sea cow) we cannot know whether gene duplications in these lineages are also enriched for tumor suppressor functions. Thus, our use of available genomes and duplication inference methods cannot address the question of convergent molecular changes underlying convergence in large bodies.

However, we do have genomes from hyrax and manatee which allows us to identify gene duplicates in these lineages and determine which gene duplication events are ancestral and derived in each lineage. For example, a gene that is duplicated in all Proboscideans but not in either manatee or hyrax will be reconstructed as duplicated in the Proboscidean stem-lineage.

One caveat to our decision to implement a cut-off of 80% when assigning ancestral copy numbers is the fact that a gene may be missing from ancient genomes due to coverage and assembly errors, and be called as “independently-duplicated” in Manatees and Elephants. While these events would underestimate the number of gene duplications that occurred in the Tethytherian stem lineage, they would ultimately not affect our conclusions that these tumor suppressors duplicated during an increase in body size in these two separate lineages. We address this in more depth in Point 6.

We acknowledge that there may be “ghost” duplication events, ie., genes that duplicated in, for example, the Tethytherian stem-lineage that are maintained in the Stellar’s sea cow genome and lost in the manatee genome; This will be reconstructed as a Proboscidean-specific gene duplication. Without genomes from multiple extinct Sirenian and Hyracoidea lineages we are not able to differentiate these alternate duplication histories. We have added a note to our Materials and methods section “Reconstruction of Ancestral Copy Numbers” to ensure this limitation of the method is clear.

While we only have genomes from extant or recently extinct species, which indeed represents a single macroevolutionary observation of increased body size/lifespan and reduced cancer risk, we identify hundreds of gene duplication events, which represents hundreds of microevolutionary observations. If there are no biases in gene duplication rates between different types of genes (and there may be) or in the probability that different types of gene duplications will reach fixation under a neutral evolutionary process (and there may be), then we should not expect to observe enrichment of a priori defined gene and pathway ontology terms among the fixed duplicate genes. That we observe the “right” pathway enrichments in the “right” lineages indicate that the hundreds of microevolutionary observations we identify support our single macroevolutionary inference.

6) Similarly, the authors observe what appears to be a number of independent duplications of tumor suppressors in African and Asian elephants: duplications that are lacking in many of the ancient genomes considered. The authors used rigorous statistical methods to correct for the fragmented nature of these ancient genomes, but it is very hard not to wonder if some of the data in Figure 4 is really not an artifact of using ancient genomes, where detecting recent gene duplications may be very difficult (several of the Asian and African elephant duplications in Figure 4 appear to be of the same genes). If these events are truly independent and not genome assembly/annotation artifacts, there is then an alternative hypothesis to propose. Thus, are the authors suggesting that there is a rapid turnover in the duplication of tumor suppressors, such that all elephants have such duplicates, but the particular duplications have short life spans and differ from species to species?

We apologize if that this figure was unclear and thank the reviewers for pointing out this problem, which we did not appreciate. We note that we are unable to determine duplication status for some genes in Proboscideans because of assembly gaps in ancient genomes; these genes appear to be independently duplicated in extant species (African Forest, African Savanah, and Asian elephants) because they are missing from ancient genomes leaving their ancestral duplication status unresolved. Thus, some of these apparently independent duplication events are because those genes are missing in the ancient genomes because of assembly gaps. We did not mean for this figure to suggest that there is a rapid turnover of tumor suppressors.

As discussed in Point 5, the absence of a gene duplicate in ancient genomes would result in our maximum likelihood method assigning the gene duplication event to the daughter lineages, in this case either Elephantidae or Sirenia. While this would underrepresent the number of genes duplicated in the Tethytheria stem lineage, we expect that this artifact would affect any duplicated gene in general, and not any particular gene or pathway; as a result, we do not expect that this would bias our outcomes in a particular direction with regards to tumor suppressor pathway enrichments.

To ensure this is clear we have redesigned Figure 4C to match the lineage coloring scheme from Figure 1, and indicate with a skull and cross bones which species are extinct. We have also added a caveat to the description of Figure 4C: “Cladogram of Afrotheria with sequenced genomes. Exemplar tumor suppressor duplicates are mapped onto lineages in which those genes are duplicated. Dots represent a duplication event of the color-coded genes. Note that we are unable to determine duplication status for some genes in Proboscideans because of assembly gaps in ancient genomes (indicated with skull and crossbones); these genes appear to be independently duplicated in extant species (African Forest, African Savanah, and Asian elephants) because they are missing from ancient genomes, biasing ancestral reconstructions of duplication status.”

7) It would be nice to see a few more comments on the manatee genome and why it does (or doesn't) show the expected patterns for the genome evolution in the face of the evolution of larger body sizes.

As we note in section “Pervasive duplication of tumor suppressor genes in Afrotheria,” 17.8% (10/56) of the pathways that were enriched in manatee (1.11-fold IBM, 0.89-log2 RICR) are related to tumor suppression. Manatee also was enriched cancer pathway percentages above background with respect to its gene set sizes, i.e., expected enrichments based on random sampling of small gene sets (shown in Figure 3B). Thus, manatee shows the expected patterns of tumor suppressor gene duplication during the evolution of larger body sizes. In addition, in the recently-added text regarding manual curations in response to Point 1, we highlight how some of the genes in Figure 4C are also highly conserved in Manatee in addition to Elephant.

8) Figures 3 and 4 would benefit from greatly expanded captions: we do not fully understand what is being illustrated in, for instance, Figure 3B-why are certain dots connected with lines? Intersections between what in the y-axis label?

We apologize if these plots were unclear. An Upset plot is a Euler diagram that shows the intersection between multiple sets, and is a way to show intersection between more than 3 sets without using Venn diagram like overlapping shapes. In Upset plots lines indicate the intersection between sets, in this case pathway terms. We have introduced a description of Upset plots in Figure 3 and a “see Figure 3 for description of upset plots” note in the caption for Figure 4.